# The Pan-African Surgical Healthcare Forum: An African qualitative consensus propagating continental national surgical healthcare policies and plans

**Barnabas Tobi Alayande**[1], **Justina O. Seyi-Olajide**[2], **Betel Amdeslassie Fenta**[1,3], **Faustin Ntirenganya**[4], **Nkeiruka Obi**[5], **Robert Riviello**[1,3], **Sabin Nsanzimana**[6], **Emmanuel M. Makasa**[7,8,9‡], **Emmanuel A. Ameh**[10‡], **Abebe Bekele**[1‡*], on behalf of the Pan-African Surgical Healthcare Forum collaborators[¶]

1 Center for Equity in Global Surgery, University of Global Health Equity, Kigali, Rwanda, 2 Department of Surgery, Lagos University Teaching Hospital, Lagos, Nigeria, 3 Program in Global Surgery and Social Change, Harvard Medical School, Boston, Massachusetts, United States of America, 4 Department of Surgery, University of Rwanda, Kigali, Rwanda, 5 Smile Train, Lagos, Nigeria, 6 Office of the Honorable Minister of Health, Ministry of Health, Kigali, Rwanda, 7 Wits-SADAC Regional Collaboration Center for Surgical Healthcare Improvement (WitsSurg), University of the Witwatersrand, Johannesburg, South Africa, 8 Center for Surgical Healthcare Research (CSHR), Lusaka, Zambia, 9 Department of Surgery, University Teaching Hospitals -Adult, Ministry of Health, Lusaka, Zambia, 10 Department of Surgery, National Hospital, Central Business District, Abuja, Nigeria

‡ EMM, EA and AB contributed equally to this work and share co-senior authors.
¶ Membership of the Pan-African Surgical Healthcare Forum collaborators is provided in the Acknowledgments.
* abekele@ughe.org

**Data Availability Statement:** Data underlying findings are available in the manuscript and supplements (specifically S4 Appendix).

## Abstract

Access to equitable, safe, affordable, timely, and quality surgical healthcare in Africa remains limited. Few African countries have surgical healthcare plans or policies. Where these exist, there are significant gaps in dissemination, funding, and implementation. A Pan-African Surgical Healthcare Forum (PASHeF) was initiated to address this. The inaugural forum was a two-day consensus conference of technocrats from African Ministries of Health hosted by the Honorable Minister for Health of Rwanda in Kigali. Through coordinated discussions, plenary sessions, working groups, and technocrat networking, they charted the path forward for national surgical healthcare policies and plans. Discussions were sparked by country experiences, and working groups focused on curated context-specific, face-validated questions. Documentation involved field notes, audio recordings, and artificial intelligence transcription. Data was coded using a constant comparative method to itemize delegates' observations, declarations, and recommendations, with member checking. A consensus statement was generated using an inclusive decision-making model. Thirty-two Ministries of Health were represented by 42 delegates who drafted and unanimously adopted the PASHeF 2023 Consensus Statement. This was a 50-point consensus addressing country commitment, leadership, financing, stakeholder mobilization, monitoring and evaluation, partnerships, and other aspects of national surgical healthcare planning in Africa. This consensus is the African roadmap and emphasizes implementation, the need

**Funding:** The authors received no specific funding for this work.

**Competing interests:** The authors have declared that no competing interests exist.

for flexibility in policy development, and current opportunities and barriers. It emphasizes that community involvement and sustainability should undergird this planning, in addition to a focus on the entire spectrum of surgical healthcare, including prevention and rehabilitation. Delegates endorsed PASHeF as an annual event with a secretariat and recommended the creation of a Pan-African Surgical Healthcare Policy monitoring system, and that issues of surgical healthcare should be escalated as an agenda item on African Union and sub-regional ministerial meetings. African nations have embraced surgical healthcare policy as an imperative on their journey towards Universal Health Coverage.

## Introduction

There are significant gaps in access to surgical healthcare across Africa [1, 2]. Although high quality surgical healthcare is a core component of Universal Health Coverage, a high burden of surgical disease, low surgical provider density, low universal health coverage, inadequate surgical infrastructure, and limited financing for surgical care are the reality for most of the fifty-five countries on the African continent [3–5]. These challenges highlight the need for realistic, locally contextualized, and progressive surgical policies which address national and regional needs. In parallel to this widespread challenge of lack of access to surgical healthcare, over 917 million Africans (76% of the continent's population) do not live under the coverage of a specific and actionable national surgical healthcare policy that charts a path for addressing the challenge [6]. There are clear gaps in policy formulation, dissemination, uptake, funding, and implementation [6]. These gaps in surgical healthcare policy are greatest in West, Central, and Southern Africa [6]. Although national surgical healthcare policies originated from Africa in response to Low- and Middle- Income Country challenges, there is lack of clarity on how African nations truly perceive the concept, process, and future of national surgical healthcare policies and planning. If surgical healthcare is truly a public health imperative, and access to surgical healthcare is indeed a fundamental human right [7, 8], then there must be discussions within and among African nations on how to translate the target of "surgical care for all by 2030" to the lived reality of the African population [1, 9].

Stemming from this need, a Pan-African Surgical Healthcare Forum was envisioned by African surgical providers, policy experts, educators, and government officials from African academic global surgery centers and countries experienced with national surgical healthcare policy and plans with the support of a non-governmental organization. The overarching aim of the forum is to bring together key stakeholders, including policymakers, clinicians, researchers, and advocates, to discuss and address the challenge of improving surgical healthcare in Africa. The forum will provide a platform for sharing innovative approaches, practical experiences, and best practices, and help with mobilization of resources to strengthen and advance surgical healthcare on the African continent.

The inaugural forum was a consensus conference [10] which sought to clearly define the direction of national surgical healthcare planning for Africa, to identify peculiarities and challenges in the planning and implementation process, with an aim of finding collective solutions to local challenges without the interference of external funders, non-governmental organization, academic institutions, and regional or multilateral organizations. African Ministries of Health were the main participants at the meeting, and it was clear from the start that discussions and decisions belonged to the participants and not the organizers, sponsors, or coordinators.

## Methods

A two-day consensus conference was held in Kigali, Rwanda on the 13 and 14 July 2023 hosted by the Republic of Rwanda Ministry of Health and championed by the Honorable Minister of Health of Rwanda. The conference comprised of an opening ceremony followed by discussions stimulated by plenary sessions, working groups, resolution drafting and reviews, and networking sessions focused on national surgical healthcare plans and policies. Delegates were invited from 53 African countries through letters to their health ministries and additional communication to Ministers representing African Member States at the 76th World Health Assembly.

### Researcher characteristics and reflexivity

A 7-person technical working group was formed with representation from the University of Witwatersrand- Southern Africa Development Community's Regional Collaboration Centre for Surgical Healthcare (WitsSurg) [11], the Center for Equity in Global Surgery, University of Global Health Equity [12], Smile Train [13], and global surgery experts from the University of Rwanda, the National Hospital Abuja, Nigeria and the Lagos University Teaching Hospital, Lagos, Nigeria (S5 Appendix). Technical group members were experienced in national surgical healthcare planning and previously contributed to the World Health Assembly Resolution WHA 68.15 on "Strengthening Emergency and Essential Surgical Care and Anaesthesia as a Component of Universal Health Coverage" [14], Ethiopia's Saving Lives Through Safe Surgery (SaLTS) [15, 16], Zambia's National Surgical, Obstetric and Anesthesia Plan (NSOAP) [17, 18], Nigeria's National Surgical, Obstetric, Anaesthesia and Nursing Plan (NSOANP) [19, 20], Rwanda's National Surgical, Obstetric and Anaesthesia Plan, and Zimbabwe's National Surgical Obstetrics and Anaesthesia Strategy [21]. A Rwanda-based logistics committee from the Ministry of Health, Rwanda, and the University of Global Health Equity organized logistics for the meeting.

### Inclusivity in global research

Members of the local communities guided the aims, were integral to the methodology and anticipated outcomes as local participants at the Pan African surgical Healthcare Forum. This was an inclusive African consensus. Additional information regarding the ethical, cultural, and scientific considerations specific to inclusivity in global research is included in the (S1 Checklist).

### Qualitative approach and research paradigm

We chose a grounded theory approach to generate explanations through an inductive, flexible, and iterative method of data collection because this was practical and would facilitate a holistic understanding of the phenomenon of national surgical healthcare plans across Africa. We used a constructivist method in interpreting participants interpretations and perspectives as these are socially constructed and context dependent. This approach also encouraged deep exploration and analysis of participants lived experiences, beliefs, and meanings attributed to national surgical planning. Constructivist methods empower participants, are collaborative and participatory, and encourage co-creation of knowledge while encouraging critical reflexivity of the researchers. A checklist on Standards for Reporting Qualitative Research is included (S1 Appendix).

### Plenaries, open discussions, and breakout sessions

Plenary sessions and experience sharing by countries already experienced in surgical plan development and implementation served as catalysts for discussion (S2 Appendix).

**Table 1. Breakout session discussion questions.**

|   | Breakout Group Discussion Focus |
|---|---|
| 1 | Are National Surgical Policies/Plans relevant to Africa? |
| 2 | Are National Surgical Policies/Plans important to your country? |
| 3 | What are the perceived advantages? |
| 4 | What are the perceived disadvantages? |
| 5 | What are the main opportunities African countries have in the development of National Surgical Policies/Plans? |
| 6 | What are the main potential barriers to the development of National Surgical Policies/Plans? |
| 7 | What are the main potential barriers to the implementation of National Surgical Policies/Plans? |
| 8 | Are the proposed steps of National Surgical Plan development appropriate? |
| 9 | Who are the major stake holders in the process of National Surgical Policy /Plan development, and how can they be brought onboard? |
| 10 | How shall African modify the steps? |
| 11 | What changes are required at the Ministries of Health level to implement significant surgical systems strengthening? |
| 12 | What should be the role of other ministries and arms of government in the design and implementation of National Surgical Policies? |
| 13 | What should Ministries of Health do within their ministry to support and sustain progress in National Surgical Plan implementation? |
| 14 | How can government mobilize and coordinate efforts of local partners to support the design and implementation of National Surgical Plans/Policies? |
| 15 | What should the roles of inter-governmental bodies in the planning and implementation of National Surgical Plans/ Policies? |
| 16 | How can ministries integrate funding of surgical and anesthesia care to other existing healthcare systems in a country? |
| 17 | What do you suggest as the way forward? |

Francophone, Lusophone, Spanish speaking, and Anglophone countries shared experiences and insights, with live translation available for all sessions. An open microphone question and answer session and moderated discussion followed plenaries with a focus on clarifying the African perspective on national surgical healthcare planning based on the preceding session. Working group breakout sessions on a wide variety of topics were also an opportunity for more intimate discussion between delegates. Moderated multi-country working groups discussed and documented answers to face-validated questions (Table 1) and plenary concerns. The face-validated questions for the working group breakout sessions were obtained and collectively refined by content experts on the technical working group and were adopted over multiple meetings with 4 iterations from March to July 2023. Following the breakout sessions, the delegate working groups brought feedback to all delegates for further discussion by the entire forum.

## Documentation and analysis

Field notes were taken during the plenary, discussion, breakout, and feedback sessions by 8 bilingual medical students with interest in global surgical care and 2 Masters in Global Health Delivery students, supervised by a postgraduate fellow in Global Surgery and the Technical Working Group. Two independent observers captured each session in detail, and their independent documentation was triangulated with documentation by the postgraduate fellow, and an administrative assistant as supervised by members of the Technical Working Group. A qualitative audit trail comprising all field notes was kept. Artificial Intelligence (Otter AI, Microsoft Word dictation) was used to capture and transcribe session content. In addition, the plenary sessions were audio-recorded by a professional audio studio for reference. The

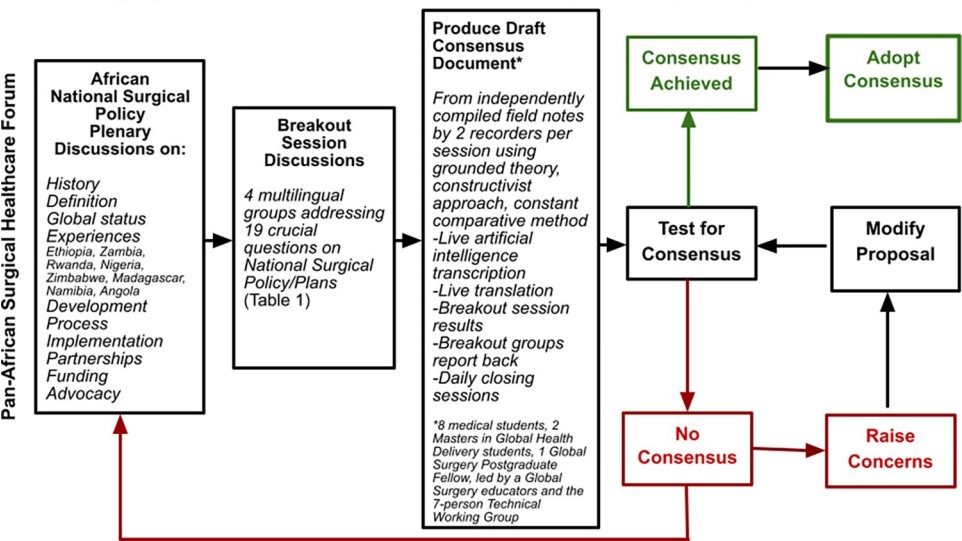

**Fig 1. Consensus document generation and adoption process.**

Technical Working Group coded the data from the recording team using a constant comparative method [22] and itemized the delegates' observations, declarations, and recommendations on each day and fed back data, analytical categories, interpretations, and conclusions to delegates at the end of each day of the forum. This member checking strengthened the data and gave delegates an opportunity for feedback on the daily outcomes of the forum [23].

## Consensus document generation and adoption process

Following feedback from delegates, daily output from the forum was compiled into a consensus statement. An agreement seeking, a priori consensus decision making process model (Fig 1) that is inclusive, collaborative, participatory, and cooperative was adopted to generate the final consensus document [24]. The criterion for including any statement in the declaration required unanimous agreement. Participants were encouraged to seek clarity, block consensus if needed, agree with reservations, or abstain with explanations. Using a visual participatory review, line-by-line reading of the draft consensus document alongside live updates based on feedback from delegates resulted in 4 iterations, followed by unanimous adoption of the PASHeF 2023 Consensus Statement [10].

## Ethics approval and informed consent

The consensus forum was determined to meet the criteria for Institutional Review Board Academic Ethics Review exemption by the University of Global Health Equity Institutional Review Board (UGHE-IRB/2023/055). Signed written informed consent was obtained from participants in the form of release forms which comprised permissions for audio and video recording, transcribing, incorporation of statements into the consensus process and photography. Forum participants received invitation letters which explained the consensus nature of the forum. During initial contact on the introductory session on the first day, the voluntary nature of audio-visual capture was verbally emphasized by organizers before written consent was obtained. No minors participated in the forum, and we did not obtain any consent from parents or guardians.

## Results

Fig 2 shows 31 countries (56% of all 55 African Union Member States) represented at PASHeF 2023 (Fig 2). There were 81 attendees at the forum, including the Honorable Minister for Health of Rwanda, and 32 African Ministries of Health (inclusive of Somaliland) were represented by 42 delegates. In addition to the Technical Working Group of 7, in attendance were 32 observers and logistics team members. A 50-point Consensus Statement resulted from the forum (Table 2). Anglophone, Francophone, Lusophone, and Spanish statements are also contained in S3 Appendix. Empirical data from detailed proceedings are contained in S4 Appendix.

## Case studies

Success stories, specific lessons, and challenges from Ethiopia (Elubabor Buno), Zambia (Christopher Chanda), Nigeria (Bitrus Oghoghorie Deborah), Zimbabwe (Shingai Nyaguse), Madagascar (Rado Razafimahatratra), Namibia (Francina Marukuavi Ngakuzevi), Rwanda

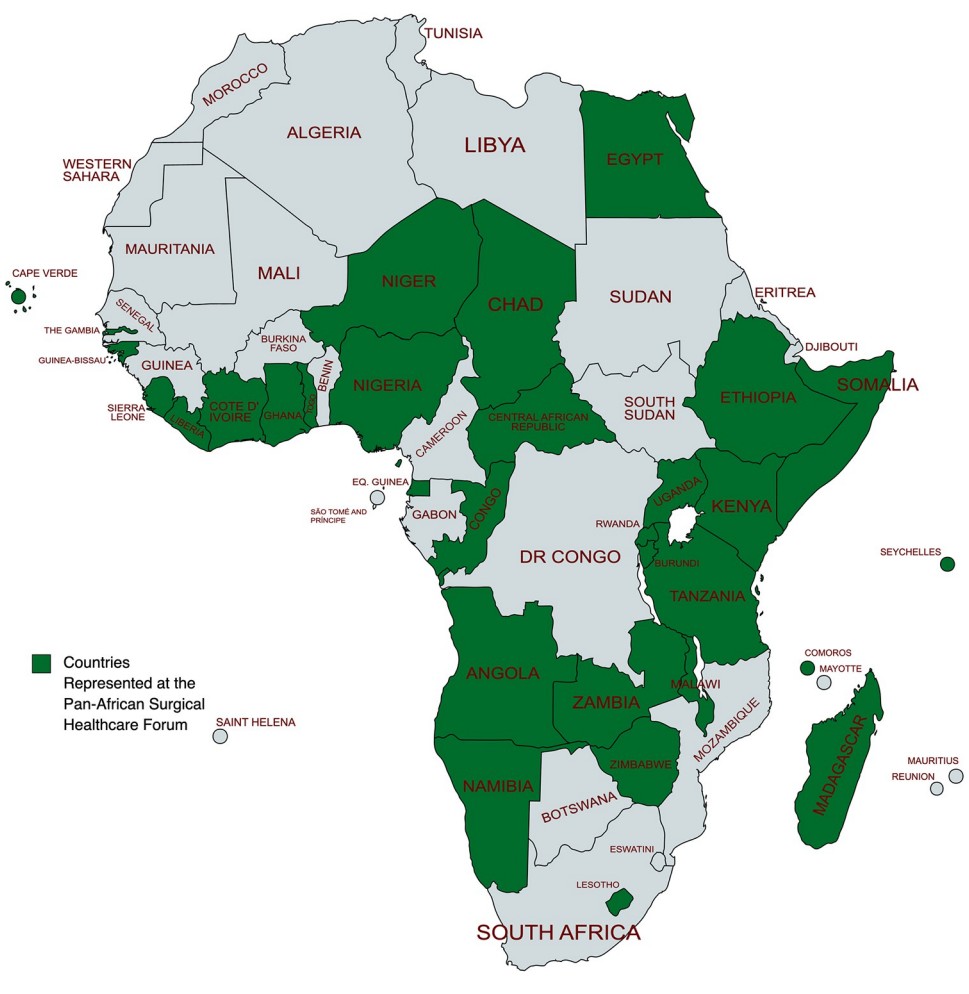

**Fig 2. African nations in attendance at the Pan-African Surgical Healthcare Forum 2023.** Created with mapchart. net. Direct link to base layer of the map at https://www.mapchart.net/africa.html.

**Table 2. Pan-African Surgical Healthcare Forum 2023 consensus statement.**

| Domain | Number | Itemized Consensus Statement |
|---|---|---|
| Relevance | 1 | Country delegates unanimously agreed that national surgical plans/policies are important to Africa. |
| | 2 | Country delegates agreed that all African countries should complete their plans/policies at the soonest. |
| | 3 | Country delegates recognized that the building blocks of surgical healthcare plans/policies can be modified to address each country's needs. |
| | 4 | There was emphasis on "not being married" to the acronym NSOAP, as it may result in lack of buy-in from some key stakeholders. |
| Importance | 5 | Country delegates agreed that surgical plan/policy implementation can lead to National Health Systems Strengthening. |
| | 6 | Country delegates urged all countries to accelerate progress towards UHC and the SDGs to meet the 2030 goals as specified in the WHA resolution 68.15. |
| | 7 | Countries need to urgently implement schemes for health insurance that covers all surgical conditions. |
| | 8 | The leadership of a "surgical plan/policy champion" would add value to the planning/policymaking and implementation. |
| Pitfalls | 9 | However, all country delegates cautioned NSOAPs alone are not the ultimate solutions to the lack of access to surgical care in the continent. |
| | 10 | Surgical plan/policy serves as the foundation, but implementation of the plan/policy is critical. |
| | 11 | Governments should commit to improve the surgical infrastructure and equipment and supply chain structure in their countries as part of plan/policy implementation. |
| National Policy Integration | 12 | Country delegates recommended surgical plans/policies should be designed to serve as integral components of the existing health care policy framework in countries. |
| Information and Data | 13 | Country delegates agreed that surgical plan/policy development and implementation should be based on local solid evidence and data (from district level to national level). |
| | 14 | They also recommended countries should strengthen and use their national health information systems and other healthcare data sources to capture surgical data. |
| | 15 | Countries should have a clear surgical care monitoring and evaluation tools and sets of indicators. |
| Opportunities | 16 | Country delegates recognized the following opportunities in surgical plan/policy development and implementation: regional partnerships for sharing of resources and expertise; existing success stories in surgical planning/policymaking in African countries; a young and energetic workforce in African countries. |
| Barriers | 17 | Country delegates identified the following barriers to surgical plan/policy development and implementation: inadequate financing for planning/policy-making and implementation; reduced political will and governmental ownership; competing priorities at a national and local level; human resource issues (lack of expertise and motivation resulting in brain drain); the systematic exclusion of surgical care in the last few decades from primary healthcare; shortage of resources; insufficient technical equipment and biomedical support; disparity between written budgets and actual financing of surgical care. |
| Proposed steps of plan development | 18 | Country delegates agreed the proposed steps of National Surgical Healthcare Policy development are appropriate. |
| | 19 | Country delegates agreed that the process should also include the community (end user) in planning, implementation, and leadership. |
| | 20 | Country delegates concurred that policies should focus on the entire spectrum of surgical care (prevention, perioperative care, and rehabilitation). |
| | 21 | Plans/policies should also emphasize capacity building, quality of care, access to care and sustainability. |
| Monitoring and Evaluation | 22 | Country delegates proposed the creation of a Pan-African/regional Surgical Healthcare Policy monitoring system composed of member states to follow up the continental progress. |
| | 23 | Monitoring and evaluation need to be aligned with Quality Assurance. |
| Ministries of Health: Leadership | 24 | Countries encouraged the creation of a dedicated surgical care leadership department/directorate at the Ministry of Health level with dedicated and qualified professionals. |
| | 25 | Members want to see surgical plans/policies institutionalized within their national healthcare systems to allow for continuity. |
| | 26 | Countries should bring stakeholder sector ministries into an inter-ministerial task force for joint leadership and supportive supervision. |
| | 27 | Country delegates recommended this plan/policy should be embraced at the highest level of government (cabinet and parliament). |
| | 28 | Ministries of Health should create a committee (Technical Working Group) or advisory council of experts to advise and monitor the implementation of the plans/policies. The leadership structure should also be reflected at the district level. |
| | 29 | The World Health Organization should be an integral part of the planning/policymaking and implementation, in close partnership with the Ministries of Health. |

*(Continued)*

**Table 2.** (Continued)

| Domain | Number | Itemized Consensus Statement |
|---|---|---|
| *Ministries of Health: Resources* | 30 | Country delegates recommended Ministries of Health should establish a budget line specific to Surgical and Anesthesia care. |
| | 31 | Budgets for surgical care should be managed at every level–Ministries of Health, Hospitals, and Health Centers. Budget should be dynamic and incremental. |
| | 32 | The national essential drug and resources list should include the appropriate surgical equipment and consumables. |
| | 33 | Countries should implement innovative ways of surgical healthcare financing. |
| | 34 | Countries should engage the private sector to mobilize resources for surgical system strengthening. |
| *Stakeholders and Partners* | 35 | Country delegates identified critical stakeholders in the process of Surgical Healthcare Plan/Policy development: including local governmental level leadership (Ministries of Health, Ministries of Finance); end users (patient groups and the community) such as women groups; the private sector and industry; professional societies; academia; NGOs; the military; WHO and other global bodies; African Union; funding agencies such as World Bank, IMF, AfDB; regional bodies such as the SADC, EAC, WAHO, ECOWAS, CEMAC and COMESA among others. |
| | 36 | Ministries of Health should strengthen their partnerships with international organizations, regional bodies, and development partners. |
| | 37 | Partnerships should be equitable and transparent. |
| *Advocacy* | 38 | Stakeholders should be engaged early in the process of Surgical Healthcare Plan/Policy development. |
| | 39 | Country delegates underscored the value of celebrities as advocates for surgical planning/policymaking. |
| | 40 | All country delegates agreed that community level advocacy is essential to promote prevention of surgical diseases. |
| | 41 | The role of the media as a positive influencer of Surgical Healthcare Plans/Policies is key for the implementation of African Surgical Healthcare Plans/Policies. |
| | 42 | Surgical systems improvement should be advanced as a public health concern. |
| *Regional Harmonization* | 43 | Country delegates recommended the following: facilitated and fast-track importation of equipment; taxation-free importation of surgical consumables and equipment; regional harmonization of training curricula; unified regional procurement system; standardization of care delivery guidelines and protocols; strengthen biomedical and infrastructure maintenance. |
| *Structure and Logistics* | 44 | PASHeF should be an annual platform for countries to share experience, request for technical exchange and support, and engage global partners. |
| | 45 | PASHeF should invite other global partners to the annual sessions. |
| | 46 | PASHeF should be hosted by alternating countries. |
| | 47 | Each African country should have a focal person represented at the PASHeF. |
| | 48 | PASHeF should develop an inclusive governance system that is representative of the whole continent. |
| | 49 | All country delegates requested this consensus document to be formally submitted to their Ministers of Health for accelerated implementation. |
| | 50 | All country delegates recommended that issues of surgical healthcare should be an agenda item on ministerial meetings, both at the African Union meetings and sub-regional meetings |

(Parfait Uwaliraye) and Angola (Carlos Alberto Pinto De Sousa) were discussed as case studies. These have been included in S4 Appendix.

## National surgical healthcare plans and policies: General concepts

Based on the large pool of documented evidence for a high burden of surgical disease, low Surgery, Anesthesia, and Obstetric provider density, and widespread lack of access to surgical care on the African continent, country delegates unanimously agreed that national surgical healthcare plans and policies are important to National Health Systems Strengthening. They observed the systematic exclusion of surgical healthcare, in the last few decades, from African Primary Healthcare discussions, and urged all African countries to accelerate progress towards UHC and the SDGs to meet the 2030 goals as specified in the WHA resolution 68.15. Country representatives emphasized the need to urgently implement schemes for health insurance that cover all surgical conditions, and that surgical systems strengthening should be advanced as a public health concern. Country delegates recommended that all African countries should

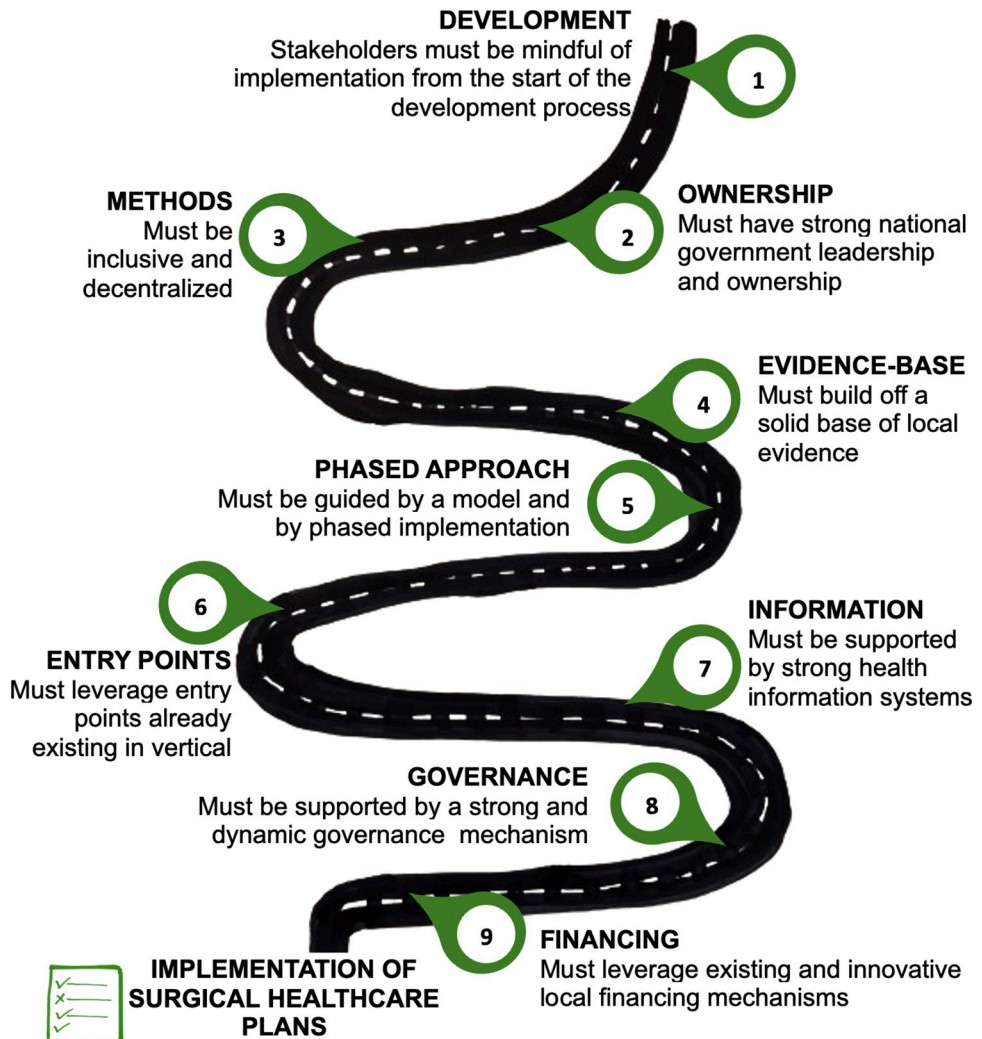

**Fig 3. The pathway to implementation of surgical healthcare policies.**

complete their plans and policies at the soonest and further that these plan and policy should be embraced at the highest level of government (Cabinet and Parliament). Fig 3 highlights the agreed roadmap to implementation of a national surgical healthcare policy (Fig 3).

However, delegates cautioned that National Surgical Healthcare Plans and Policies alone are not the ultimate solutions to the lack of access to surgical care on the continent. While the presence of a surgical healthcare plan or policy document serves as a solid foundation, implementation is critical for results. African Member States also want to see surgical plans and policies institutionalized within the national healthcare systems, beyond being the project of interested political actors, to allow for continuity/sustainability.

## Opportunities and barriers to surgical healthcare planning and implementation

Country delegates recognized regional partnerships for sharing of resources and expertise, existing success stories in surgical planning and policymaking in African countries, and a young and energetic workforce in African countries as significant opportunities in surgical

policy development and implementation. However, delegates identified inadequate financing for policy development and implementation, reduced political will, inadequate governmental ownership, and competing priorities at a national and local levels as barriers to surgical policy development and implementation. They also noted human resources for health challenges (including lack of expertise and poor motivation resulting in brain drain, shortage of surgical resources, insufficient technical equipment, insufficient biomedical support, and the disparity between written budgets and actual financing of surgical healthcare as barriers to policy development and implementation.

Discussions about vulnerable populations such as pediatrics stemmed from national surgical planning case studies from Nigeria and Madagascar (see S4 Appendix. Proceedings and empirical data), and participants acknowledged the high proportion of children in Africa as a peculiar consideration in planning for surgical care on the continent.

## Partnerships for surgical healthcare planning and implementation

Country delegates identified critical stakeholders in the process of Surgical Healthcare Plan and Policy development: including local governmental level leadership (MoH, Ministries of Finance); end users (patient groups and the community) such as women groups; the private sector and industry; professional societies; academia; non-governmental organizations; the military; the World Health Organization and other global bodies; African Union; funding agencies such as World Bank, International Monetary Fund, the African Development Bank; regional bodies such as the South African Development Community, the East African Community, the West African Health Organization, the Economic Community of West African States, Communauté Économique et Monétaire de L'afrique Centrale and The Common Market for Eastern and Southern Africa, among others. Ministries of Health should strengthen their partnerships with international organizations, regional bodies, and development partners. Stakeholders should be engaged early in the process of Surgical Healthcare Plan and Policy development. However, these partnerships should be equitable and transparent.

The leadership of a "Surgical healthcare plan or policy champion" in each country would add value to the planning and implementation. The role of the media as a positive influencer of Surgical healthcare policies is key for the implementation of African surgical healthcare policies.

## The process of surgical healthcare planning

Country delegates agreed that the proposed steps of National Surgical Healthcare Policy development in the United Nations Institute for Training and Research National Surgical planning manual are appropriate[25, 26]. However, they noted that not enough emphasis has been placed on the role of the community within the framework of the roadmap to implementation. Country delegates agreed that the process should also include the community (end user) in planning, implementation, and leadership (Fig 4). Delegates insisted that beyond a focus on management of surgical conditions and provision of surgical healthcare for established surgically treatable disease, national surgical healthcare policies should incorporate community-based prevention of surgical diseases and rehabilitation. All country delegates agreed that community level advocacy is essential to promote prevention of surgical diseases.

In addition, for African countries, policies should emphasize capacity building, quality of care, access to care and sustainability. The framework should include an emphasis on the ethos of sustainability (Fig 4) in various dimensions. Sustainability in this sense refers to "a state in which disadvantaged communities or developing countries can address health challenges and provide quality, equitable healthcare to their populations with limited reliance on external

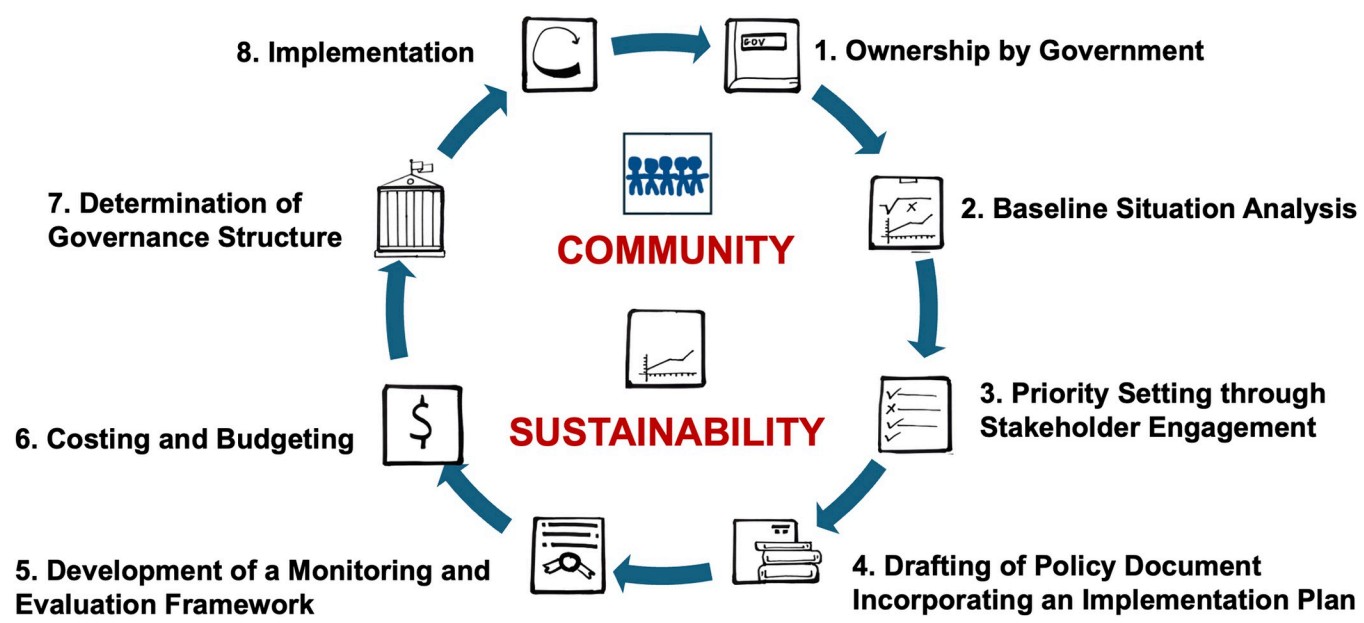

**Fig 4. Expanded surgical planning theoretical framework: Development of national surgical healthcare plans for Africa should incorporate a strong community and sustainability ethos.**

support" [27]. This includes the ability of the national surgical healthcare plan to continue to function effectively, for the foreseeable future, with high utilization, integrated into available health care services, with strong community ownership using resources mobilized by the local community and government. Environmental sustainability should also be part of the sustainability ethos undergirding national surgical healthcare planning.

The road-map to implementation of a national surgical healthcare policy starts from its development process and should be undergirded by strong community involvement and focus and sustainability in every dimension.

Country delegates insisted that the process in Africa should include the community (end user) in planning, implementation, and leadership, and should emphasize the "demand side" by addressing barriers hindering community access to surgical health services. Policies should focus on the entire spectrum of surgical care (with continuity of care), including prevention, perioperative care, and rehabilitation. They noted that prevention and rehabilitation have been relatively neglected emphases so far in the NSOAP planning process (Fig 5).

Country delegates agreed that plan and policy development and implementation should be based on local solid evidence and data (from district level to national level) and recommended that countries should strengthen and use their national health information systems and other healthcare data sources to capture disaggregated surgical data. A Ministry of Health based multi-stakeholder technical working group that supports an inter-ministerial task force, involving stakeholder sector ministries (including ministries of finance) should be inaugurated for surgical healthcare planning to encourage for joint leadership and supportive supervision. It was further recommended that the WHO should be an integral part of the planning/policy-making and implementation, in close partnership with the Ministries of Health.

## Financing of planning and implementation

Country delegates recommended that Ministries of Health should establish a budget line specific to surgical and anesthesia healthcare. These budgets should be managed at every level-

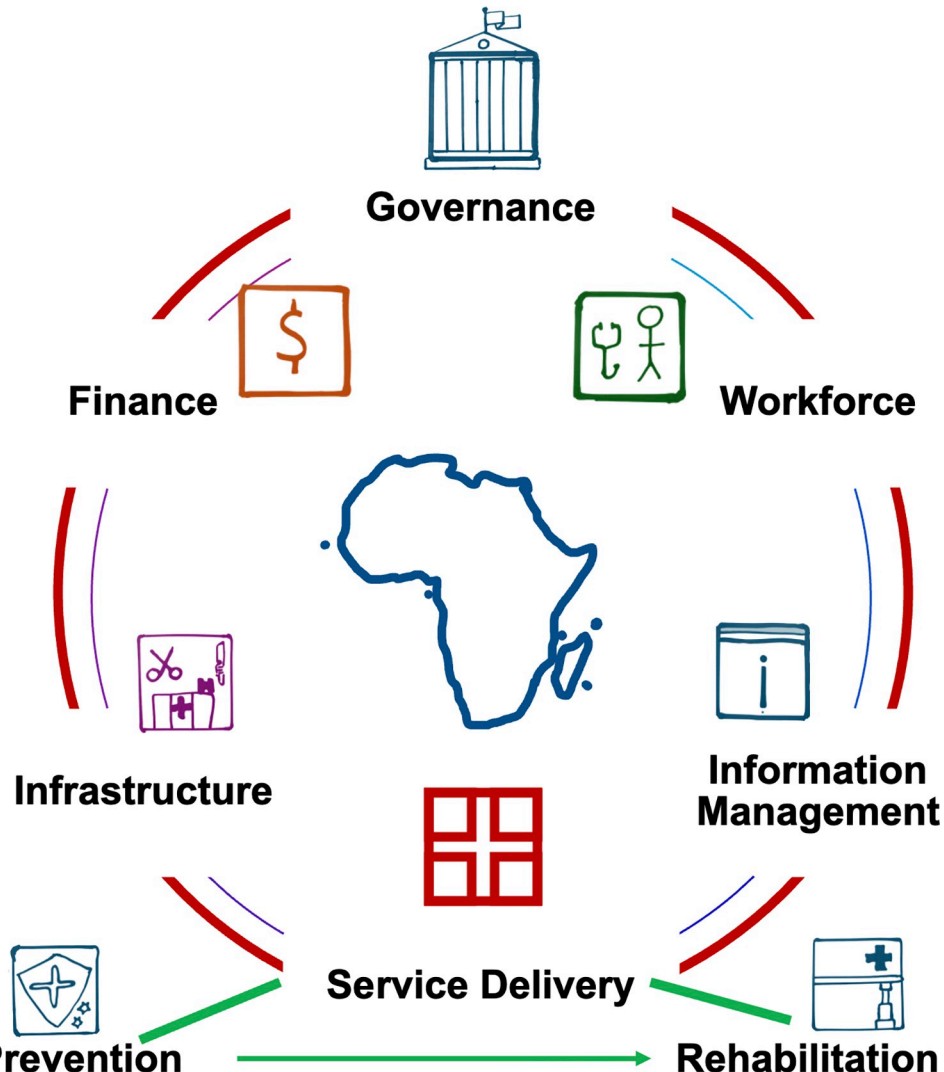

**Fig 5. Pillars of national surgical policy: Emphasis of African plans should be on the entire spectrum of service delivery from surgical disease prevention to rehabilitation.**

including Ministry, Hospital, and Health Center levels. They should be dynamic and incremental. Countries should implement innovative surgical healthcare financing including Social Health Insurance, Motor Vehicle Insurance, Bank Transaction Tax, Non-Communicable Disease related Tax, and in view of Africa's rapidly expanding mobile phone penetration (a continental average of 85%), phone airtime purchase taxes can be dedicated to surgical healthcare. Countries should engage the private sector to mobilize resources for surgical system strengthening.

## Integration of surgical policy into existing health care policy

Country delegates recommended that surgical healthcare plans and policies should be designed to serve as integral components of the existing health care policy framework in countries. Participants posited that surgical healthcare specific policies are best practice, however, they should speak to other healthcare plans. They raised concerns about the challenge of

acceptance of distinct national surgical healthcare policy in certain countries, as there might be other competing policies already in place. Participants recognize that duplication of plans and policies might waste scarce resources and recommend adequate consideration by countries to find ways to best place or integrate surgical healthcare plans within existing frameworks.

## Monitoring and evaluation of plan/policy implementation

Delegates agreed that countries should have a clear set of surgical healthcare monitoring and evaluation tools, and well-defined indicators. They insisted that monitoring and evaluation of surgical healthcare plans needs to be aligned with Quality Assurance. Countries encouraged the creation of a dedicated surgical healthcare leadership department or directorate at the Ministry of Health level with dedicated and qualified professionals. A National Surgical Planning desk office was deemed inadequate at the level of African Ministries of Health. They further recommended that Ministries of Health should create a Technical Working Group or advisory council of experts to advise and monitor the implementation of the plans and policies. The leadership structure should also be replicated at sub-national levels, down to the district level.

Country delegates stated that internal monitoring and evaluation within countries was not sufficient and proposed regional action with a peer monitoring and evaluation system. They proposed the creation of a Pan-African/regional Surgical Healthcare Policy monitoring system composed of member states to follow up continental progress in surgical care. They also suggested that monitoring and evaluation should be aligned with Quality Assurance. While monitoring and evaluation was seen as more focused on indicators and process or outcome measures, quality assurance was seen as more proactive and less focused on reprimands while ensuring that national surgical healthcare programs meet the needs and expectations of population throughout the duration of the national surgical plan implementation. Monitoring and evaluation could thus be improved by regional peer monitoring systems, strengthening data collection systems and establishing dedicated specialized units or directorates for surgical healthcare Monitoring and Evaluation. Mid-term evaluation also need to be carried out as in the Rwanda case study.

## Local flexibility in adaptation

Delegates recognized that the building blocks of the plan or policy can be modified to address each country's needs. In terms of nomenclature, there was emphasis on "not being married" to the acronym NSOAP to denote national surgical policies and plans, as this may result in lack of buy-in from some key stakeholders. The term "national surgical healthcare plan or policy" was adopted to describe the national strategic planning document for surgical care (which is not specific to surgery as a specialty, but inclusive of surgery, anesthesia, nursing care, obstetrics, and allied healthcare that contribute to surgical care) in domains like workforce, information management, governance, service delivery, infrastructure, and finance. There was concern that using the term "surgical" alone does not capture the entire spectrum intended, as it can often be perceived to refer to surgeons, however, the inclusion of every single entity that truly contributes to holistic surgical healthcare may also be perceived as divisive and contribute to exclusion of certain actors. The term "surgical healthcare" was viewed as more inclusive and encompassing.

## Advancing continental implementation of national surgical healthcare plans

Country delegates recommended the following: facilitated and fast-track importation of equipment; taxation-free importation of surgical consumables and equipment; regional

harmonization of training curricula; unified regional procurement system; standardization of care delivery guidelines and protocols; strengthened biomedical and infrastructure maintenance. Governments across Africa should commit to improving surgical healthcare infrastructure, equipment, and supply chain structure in their countries as part of plan and policy implementation. National essential drug and resource lists should include the appropriate surgical equipment and consumables.

Country delegates proposed the creation of a Pan-African and regional Surgical Healthcare Policy monitoring system composed of Member States to follow up on the continental progress. They agreed that PASHeF should be an annual platform for countries to share experience, request technical exchange and support, and engage global partners, and should be hosted on a rotational basis by alternating countries. Delegates decided that each African country should have a focal person represented at the PASHeF and that an inclusive governance system that is representative of the whole continent should be developed.

All country delegates requested that the consensus document should be formally submitted to their Ministers of Health for accelerated implementation and recommended that issues of surgical healthcare should be an agenda item at ministerial meetings, both at African Union and sub-regional meetings.

## Discussion

African countries and experts have played an active role in shaping global surgical healthcare policies and strategies. From involvement in the seminal Lancet Commission on Global Surgery to the intergovernmental negotiations that culminated in the adoption of the WHO resolution WHA68.15 on Strengthening Surgical Care and Anaesthesia as a component of Universal Health Coverage, the follow-up Decision WHA70(22), and the recent WH76.2 "ECO resolution", the contributions of African nation states are evident [14, 28, 29]. While African countries and experts have made significant strides in improving surgical healthcare, there is still much work to be done to address the significant gaps in access to safe, timely, and affordable surgical healthcare on the continent. Continued investment and collaboration among African countries, experts, and global partners are needed to ensure that all Africans have access to the surgical healthcare they need. This calls for the creation of an appropriate Africa-initiated and Africa-led multi-stakeholder platform where Member States, working with their partners, can begin to collaborate and coordinate Pan-African initiatives, mobilize the needed resources for an Africa-wide surgical response, share best practice, and learn lessons from each other.

The implications of and potential impacts of the outcome of the PASHeF forum are far reaching. African countries recognize the importance of surgical healthcare plans and policies to National Health Systems Strengthening and their role in achieving UHC and the SDGs [14, 30, 31], and emphasized the need for phased sub-national implementation [30]. One key emphasis for countries was on early engagement of stakeholders and local, regional, and international partnerships [32]—including with the media and political champions. Novel additions to the established national surgical policy framework [25] were an emphasis on the role of the community, community-based prevention of surgical disease, and community level advocacy [33, 34]. Sustainability in all its dimensions (including the dimensions of local ownership, continuity beyond current political interests, and environmental sustainability) is also an essential consideration for African surgical plans [27, 35].

National surgical healthcare planning and policy often traditionally emphasizes what delegates described as "the supply side"- provision of hospitals, increasing the workforce, and providing infrastructure. Ministries of Health have now suggested that African national surgical

healthcare policies should equally emphasize the "demand side" and address barriers to access to surgical infrastructure. This will ensure that the entire spectrum of surgical care from prevention to rehabilitation is emphasized by new national surgical healthcare policies. Also, as exemplified by the Nigerian plan, the high proportion of a vulnerable pediatric population needs to be factored into the approach to surgical healthcare planning [6, 19].

The PASHeF forum also has an implication for financing of the surgical healthcare ecosystem. Financing the implementation of national surgical healthcare policies and plans has been a significant challenge. African ministries of health have suggested dynamic, incremental, and creative financing, alongside mobilization of the private sector. This suggests that Ministries of Finance should be an integral part of surgical healthcare policy formulation as early stakeholders in planning, and not involved as an afterthought. The role of institutions like the African Development Bank group, which exist to spur sustainable economic development and social progress in its regional Member States, need to be expanded to include surgical healthcare. Financial assessments of the phased implementation of surgical healthcare policies and plans and a value proposition to African leaders is critical to attracting such funding [36].

Finally, peer accountability that results from such a forum is crucial to benchmarking the process of policy and plan development and implication among African nations with a view to providing surgical healthcare quality improvement. Indicators can be collaboratively assessed, and collective efforts can be made to strengthen surgical healthcare with joint resources. This has successfully been demonstrated in the SADC region through the University of Witwatersrand- Southern Africa Development Community's Regional Collaboration Centre for Surgical Healthcare (WitSSurg) and can be implemented on a larger scale.

Several lessons have been learned from this process which are applicable to similar resource limited settings. Firstly, creating peer review and accountability platforms at regional and continent-wide levels is possible. When these are organic and envisioned without external influence, it engenders a sense of ownership, and the process can be trusted to reflect local priorities, feasible methods, and contextual results. This PASHeF forum also demonstrates that the role of academics and academic institutions in promoting surgical healthcare can extend beyond educating surgical care providers. Beyond the vertical medical mission trips typical of several NGOs, sponsorship for this forum has shown that such institutions can be involved in surgical systems strengthening by supporting more locally led, horizontal, capacity building programs. Academic resources (staff, stuff, space, systems) can be deployed to support policy-strengthening fora. The role of a ministerial champion should not be ignored, as demonstrated by the role played by the Rwandan Ministry of Health in championing this collaborating with other national ministries of health. Advances in surgical healthcare policy and planning in other regions like the Western Pacific has also clearly demonstrated the advantage of this championing role played by a minister of Health. Finally, while consensus is not easy to achieve because of individual national interests, health, and specifically surgical healthcare is a universal rallying point for African nations.

## Limitations

Limitations include the gaps in involvement from Northern Africa, with only Egypt contributing to the consensus. This consensus may therefore not reflect the nuances of north Africa due to limited involvement from the region in the inaugural PASHeF. In addition, while participants indicated that countries should have clear surgical care monitoring and evaluation tools and sets of indicators, and that peer accountability based on these is crucial, there was no attempt at pursuing consensus geared at any specific selected regional indicators. Identifying cross-cutting indicators will be a crucial next step in evaluating systems and will be the focus of

future discussions. Countries envision that specific indicators can be collaboratively assessed for peer accountability, and this will be the focus of future discussions for PASHeF. While discussion on the role of academic and other partners came up during plenary and was highlighted in consensus statement 35, further discussion is necessary given their crucial role in global surgery and PASHeF consensus implementation. For instance, the involvement of established African regional collegiate surgical training programs (particularly the West African College of Surgeons, the College of Surgeons of East Central and Southern Africa, national postgraduate medical colleges of surgery) and university surgical training programs as stakeholders will aid in enhancing regional harmonization and can serve as academic think-tanks for Ministries of Health. They are key avenues through which countries can reach appropriate doctor- population ratios stated in their surgical plans, channels for financial aid, and can provide for monitoring and evaluation and quality assurance of policies. The means of engagement and onboarding of partners with PASHeF must be outlined and discussed by member states in the future.

## Conclusions

African nations have embraced surgical healthcare policy/planning as an imperative on the journey towards universal health care. Ministries of health have called for a broader focus on community involvement and sustainability as the core of the framework of national surgical healthcare planning, along with an emphasis on neglected aspects along the spectrum of surgical care provision- namely prevention and rehabilitation. The Pan-Africal Surgical Healthcare Forum (PASHeF), to be governed by Ministries of Member States is well positioned to provide continental mobilization, coordination, accountability, and quality improvement for surgical healthcare plans/policy and will serve as a platform for technical exchange, support, and engagement of local, regional, and global partners.

## Supporting information

**S1 Checklist. Inclusivity in global research.**
(PDF)

**S1 Appendix. Standards for reporting qualitative research.**
(PDF)

**S2 Appendix. Pan-African Surgical Healthcare Forum plenary sessions and facilitators.**
(PDF)

**S3 Appendix. English, French, Portuguese, and Spanish PASHeF 2023 consensus statement.**
(PDF)

**S4 Appendix. Proceedings and empirical data.**
(PDF)

**S5 Appendix. Reflexivity statement.**
(PDF)

## Acknowledgments

### Collaborating Authorship

The Pan African Surgical Healthcare Forum collaborators include:

Ali Said Mohamed Chafiou (Comoros), Amanda Rurangwa (Rwanda), Antoine Doui Doumgba (Central African Republic), Antonio Martin Elo Obono Elo (Equatorial Guinea), Augustus Garlet Quiah (Liberia), Ayan Abdi Harare (Somaliland), Bi Irie Laurent Toa (Cote d'ivore), Bonite Havyarimana (Burundi), Caroline Mayengo Damian (Tanzania), Carlos Alberto Pinto De Sousa (Angola), Charles Olaro (Uganda), Christopher Chanda (Zambia), Cruz Matete (Angola), Djamila Cavaleiro Principe (Angola), Beula Igiraneza (Rwanda), Carloz Zeca (Angola), Derrick Niyonkuru (Rwanda), Danny Thomas Louange (Seychelles), David Ooko Soti (Kenya), Deborah Alheri Bitrus Oghoghorie (Nigeria), Doles Hamza Sama (Togo), Eden Gatesi (Rwanda), Elubabor Buno Teko (Ethiopia), Engrácia Mouzinho (Angola), Francina Tjituka (Namibia), Gloria Nishimwe (Rwanda), Heritier Mfura (Rwanda), Ilda Jeremias (Angola), Jane Nginge (Kenya), Joselyne Nzisabira (Rwanda), Jules Iradukunda (Rwanda), Jûlîo Carvalho (Angola), Joao Rufino Ismael Lopes Jandy (Guinea Bissau), John Nkrumah Mills (Ghana), Jean Raoul Chocolat (Congo), Judith Mkwaila (Malawi), Julius Almeida (Angola), Kedest Mathewos (Ethiopia), Lieketseng Arcilia Petlane (Lesotho), Marlene Muhongerwa (Rwanda), Mohamed Safwat Abohleka Sherif (Egypt), Mohammed Tareq Orfali (Rwanda), Mustapha Sundifu Kabba (Sierra Leone), Ngaringuem Olivier (Chad), Olivier Mbarushimana (Rwanda), Omar Mohamed Abdinor (Somalia), Osman Liban Yusuf (Somalia), Ousseini Elhadji Adakal (Niger), Parfait Uwaliraye (Rwanda), Pierrette Ngutete Mukundwa (Rwanda), Rado Razafimahatratra (Madagascar), Ayan Abdi Harare (Somalia), Sam Livingstone (Liberia), Shingai Audrey Nyaguse Chiurunge (Zimbabwe), Solange Nakure (Rwanda), Tito Livio Ramos Rodrigues (Cape Verde), Tomás Cassinda (Angola).

We acknowledge the leadership of the Ministry of Health of Rwanda, and the Honourable Minister of Health, Dr Sabin Nsanzimana MD, PhD, who convened and hosted the forum, and the role of the PASHeF Technical and Logistics Committees. We also acknowledge Medical Students, Intare Surgical Society, and Masters in Global Health Delivery learners at UGHE who supported logistics and contributed to data collection.

## Author Contributions

**Conceptualization:** Barnabas Tobi Alayande, Justina O. Seyi-Olajide, Emmanuel M. Makasa, Emmanuel A. Ameh, Abebe Bekele.

**Data curation:** Barnabas Tobi Alayande, Justina O. Seyi-Olajide, Betel Amdeslassie Fenta, Faustin Ntirenganya, Nkeiruka Obi, Robert Riviello, Emmanuel A. Ameh, Abebe Bekele.

**Formal analysis:** Barnabas Tobi Alayande, Justina O. Seyi-Olajide, Betel Amdeslassie Fenta, Faustin Ntirenganya, Nkeiruka Obi, Robert Riviello, Emmanuel M. Makasa, Emmanuel A. Ameh, Abebe Bekele.

**Investigation:** Barnabas Tobi Alayande, Justina O. Seyi-Olajide, Betel Amdeslassie Fenta, Nkeiruka Obi, Robert Riviello, Emmanuel M. Makasa, Emmanuel A. Ameh, Abebe Bekele.

**Methodology:** Barnabas Tobi Alayande, Faustin Ntirenganya, Nkeiruka Obi, Robert Riviello, Emmanuel M. Makasa, Emmanuel A. Ameh, Abebe Bekele.

**Project administration:** Barnabas Tobi Alayande, Emmanuel M. Makasa, Emmanuel A. Ameh.

**Resources:** Nkeiruka Obi, Sabin Nsanzimana, Abebe Bekele.

**Supervision:** Barnabas Tobi Alayande, Sabin Nsanzimana, Emmanuel A. Ameh, Abebe Bekele.

**Validation:** Barnabas Tobi Alayande, Justina O. Seyi-Olajide, Betel Amdeslassie Fenta, Faustin Ntirenganya, Robert Riviello, Emmanuel M. Makasa, Emmanuel A. Ameh, Abebe Bekele.

**Visualization:** Barnabas Tobi Alayande.

**Writing – original draft:** Barnabas Tobi Alayande.

**Writing – review & editing:** Barnabas Tobi Alayande, Justina O. Seyi-Olajide, Betel Amdeslassie Fenta, Faustin Ntirenganya, Nkeiruka Obi, Robert Riviello, Sabin Nsanzimana, Emmanuel M. Makasa, Emmanuel A. Ameh, Abebe Bekele.

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
