## [Decision Letter · Decision Letter 0]

19 Jul 2024

PGPH-D-24-00732

The Pan-African Surgical Healthcare Forum: An African Qualitative Consensus Propagating Continental National Surgical Healthcare Policies and Plans

Dear Dr. Alayande,

Thank you for submitting your manuscript to PLOS Global Public Health. I apologize for the delay in getting this review back to you. After careful consideration, we feel that it has merit and would like to publish it in PLOS Global Public Health. However, the reviewers had some minor suggestions for improvement. We therefore invite you to submit a revised version of the manuscript that addresses the points raised during the review process.

We look forward to receiving your revised manuscript.

Kind regards,

Shahrzad Joharifard

Academic Editor

Journal Requirements:

Reviewers' comments:

Reviewer's Responses to Questions

**Comments to the Author**

1. Does this manuscript meet PLOS Global Public Health’s publication criteria? Is the manuscript technically sound, and do the data support the conclusions? The manuscript must describe methodologically and ethically rigorous research with conclusions that are appropriately drawn based on the data presented.

Reviewer #1: Yes

Reviewer #2: Yes

2. Has the statistical analysis been performed appropriately and rigorously?

Reviewer #1: N/A

Reviewer #2: I don't know

3. Have the authors made all data underlying the findings in their manuscript fully available (please refer to the Data Availability Statement at the start of the manuscript PDF file)?

Reviewer #1: Yes

Reviewer #2: Yes

4. Is the manuscript presented in an intelligible fashion and written in standard English?

Reviewer #1: Yes

Reviewer #2: Yes

5. Review Comments to the Author

Reviewer #1: I would like to congratulate the authors on an outstanding report on this effort.

I have a few questions for further clarification-

Did any discussion of vulnerable populations such as pediatrics come up or was the conversation broader?

Indicators were noted very generally - where there any comments on what LCoGS or other indicators may be considered? Again the discussion may not have been in this detail but surely an important next step will be evaluating systems.

Were there any specific lessons/success stories or challenges from any participating country that came up during the meeting? It would be useful to have these as small case studies or included in the report in some fashion.

In the M&E section on NSOAPs, why do African MOHs feel these are inadequate? What do propose differently? How would that be better?

Line 507, I'm not sure I understand what "supporting similar internally led processes" refers to. Also in the next sentence, there is reference to academic resources but one usually would not consider "staff, stuff, space, systems" academic resources. These two minor points should be considered.

Lastly there are few scattered punctuation errors - missing apostrophes and lack of capitalization etc.

Thank you for the opportunity to review this work and give constructive feedback. Congrats once again to the authors for a truly inspiring effort.

Reviewer #2: 1. With the use of the constructivist method, use of 'key words' in the coding to facilitate comparative analysis of the observations and declarations would be important to denote in the Appendix. This was unclear in Appendix 4.

2. Perhaps as a limitation, or an opportunity for further discussion, involvement of the established collegiate/ university surgical training programs as stakeholders may aid in percolating global surgery, as well as enhancing regional harmonisation. In conjunction with the respective Ministries of Health, these would serve as valuable avenues through which the respective countries would reach their doctor- population ratios stated in the NSOAPs, or facilitate channels for financial aid, act as centres for monitoring and evaluation of the policies once they would be rolled out. These already being present in some of the mentioned countries, should have representation/ be mentioned.

6. PLOS authors have the option to publish the peer review history of their article (what does this mean?). If published, this will include your full peer review and any attached files.

**Do you want your identity to be public for this peer review?** For information about this choice, including consent withdrawal, please see our Privacy Policy.

Reviewer #1: No

Reviewer #2: No

---

## [Editor Report · Decision Letter 1]

23 Sep 2024

The Pan-African Surgical Healthcare Forum: An African Qualitative Consensus Propagating Continental National Surgical Healthcare Policies and Plans

PGPH-D-24-00732R1

Dear Dr. Alayande,

We are pleased to inform you that your manuscript 'The Pan-African Surgical Healthcare Forum: An African Qualitative Consensus Propagating Continental National Surgical Healthcare Policies and Plans' has been provisionally accepted for publication in PLOS Global Public Health.

Best regards,

Shahrzad Joharifard MD MPH FRCSC

Academic Editor

Reviewer Comments (if any, and for reference):

Thank you for your patience with the editorial process and for making requested changes. Congratulations on your manuscript.